# Digital Twin Applications in 3D Concrete Printing

Yuxin Wang [1], Farhad Aslani [1,2,3,*] , Arcady Dyskin [1,2] and Elena Pasternak [1,2]

[1] School of Engineering, University of Western Australia, Perth, WA 6009, Australia
[2] UWA International Space Centre, University of Western Australia, Perth, WA 6009, Australia
[3] School of Engineering, Edith Cowan University, Perth, WA 6027, Australia
*  Correspondence: farhad.aslani@uwa.edu.au

**Abstract:** The benefits of 3D concrete printing (3DCP) include reducing construction time and costs, providing design freedom, and being environmentally friendly. This technology is expected to be effective in addressing the global house shortage. This review highlights the main 3DCP applications and four critical challenges. It is proposed to combine 3D concrete printing with Digital Twin (DT) technology to meet the challenges the 3DCP faces and improve quality and sustainability. This paper provides a critical review of research into the application of DT technology in 3DCP, categorize the applications and directions proposed according to different lifecycles, and explore the possibility of incorporating them into existing 3DCP systems. A comprehensive roadmap was proposed to detail how DT can be used at different lifecycle stages to optimize and address the four main challenges of 3DCP, providing directions and ideas for further research.

**Keywords:** 3D printing; 3DCP; digital twin; construction industry

## 1. Introduction

Global housing shortage, the effects of global climate change, and rising labor costs due to a shortage of skilled workers are driving the construction industry to question traditional construction methods and push the limits of innovation. In this quest, additive manufacturing (AM) technology (in particular 3D concrete printing) is gaining ground in the construction industry to respond to the urgent demand for acceleration and mechanization [1].

The method of automating construction by reading digital models of buildings will undoubtedly blaze a new path for the construction industry to achieve cost reduction [2], environmental protection [3], and realize complex shaped designs [4]. Additive manufacturing, such as 3D concrete printing, is well suited for adopting digital technologies and achieving automating construction. The method of 3D concrete printing is also suitable for creating structural elements with complex geometries, which allows building structures of non-traditional architecture [5]. In addition, this technology can significantly improve sustainability and reduce energy consumption, with reduced construction time and waste [6], and use short fiber reinforced filaments from recycled plastics and fibers (such as polypropylene and basalt) as a printing material [7]. Despite existing beliefs that 3D printing is a difficult alternative to traditional construction methods, there are cases when 3D printing can provide an extremely effective solution [5]. In addition to its ability to create complex shapes, it can also be effective in response to disasters [4]. Whenever a natural disaster such as a hurricane or earthquake destroys infrastructure and leaves people homeless, 3D concrete printing can be used to quickly rebuild bridges, roads, and houses. Finally, based on its low cost and high efficiency, it can be a practical option for social housing projects [8]. The past few years have seen a proliferation of 3D-printed buildings, including administrative buildings in Dubai, residential houses in various European countries [9], and apartments and bridges in China [10]. The method of 3D printing even underpins NASA's proposals for a Mars habitat [4]. This shows that this new technology, which

previously seemed rather far-fetched and fantastical, is rapidly developing. However, like any other innovation, it has a long way to go before it becomes a viable, sustainable, and widely used technology [11]. Material limitations, scale limitations, high initial investment, lack of competent personnel, and environmental sensitivities all pose challenges to the development of 3D concrete printing in the construction industry [12,13].

Recently, a new construction technology of mortarless construction based on topological interlocking producing demountable structures is emerging, thus reducing construction time and waste as well as being capable of the reduction of $CO_2$ print by reducing cement consumption [14–16]. Topological interlocking can also be effective in extraterrestrial construction [17,18]. As topological interlocking is based on blocks of complex shapes, Figure 1, 3D concrete printing can be a technology of choice in manufacturing topological interlocking blocks [19]. Considering that the structure needs special edge, corner, lintel, etc., blocks, digital technology could be incorporated to improve design and buildability.

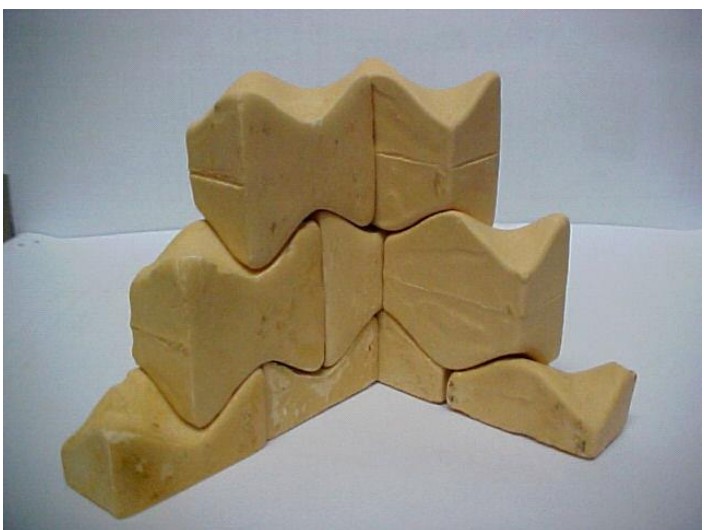

**Figure 1.** Topological interlocking shapes and structure "Reprinted/adapted with permission from Ref. [15]. Copyright 2015, Dyskin et al."

Efficient construction requires proper planning synchronization and optimization of all technological and logistic elements of construction. Digital twin (DT) technology, an analog-based planning and optimization concept, has begun to make its mark on the construction industry in recent years [20]. DT could facilitate the communication of information and the monitoring and optimization of physical entities by enabling the seamless transfer of data between physical and virtual worlds [21]. It is expected to have the potential to minimize the challenges faced by 3D printing technology during different phases.

Research related to the application of digital twin technology to 3DCP is still in its infancy. Some theoretical considerations and experiments have been conducted by the construction industry to explore the potential directions and benefits that digital twin technology can bring to 3D concrete printing. However, these studies are fragmented. Therefore, a critical review of the current state of 3D concrete printing and the development of current DT applications in it is needed. The aim of this paper is to (1) review the benefits and applications of 3D printing in the construction industry; (2) review the barriers and challenges in the application of 3D concrete printing; (3) review the current status and benefits of digital twin technology in 3D concrete printing; and (4) discuss the challenges of using digital twin technology in 3DCP and the direction of its application, and hopefully, by providing potential directions and solutions, it can be better utilized in the future to develop 3DCP.

## 2. Methodology

Given the wide coverage of scientific publications and the good performance of Scopus [22], this paper mainly uses it for the initial search of the literature. The search was also supplemented by other databases such as ScienceDirect and Google Scholar.

As the aim of this study was to review the current state of 3D concrete printing in the construction industry and the use of digital twin technology in 3D concrete printing, the search was conducted in two stages. The first stage was to review 3D concrete printing, so the keywords were divided into two main parts, the first including "3D concrete printing" or "3DCP", while the second part included "construction", "construction industry", "construction engineering", or "AEC". As the current 3DCP status is to be explored, the data range was set ranging from 2018 to 2022. A total of 296 documents were searched and then filtered according to their abstracts. In addition, some of the important and highly relevant references mentioned in the selected papers were also reviewed carefully as a supplement.

The second stage (Figure 2) was to review digital twins in 3D concrete printing and the search string also consisted of two parts. The first part includes keywords related to "digital twins", "virtual twins" or "DT". The second part includes keywords such as "3D concrete printing" or "3DCP". Only one publication was retrieved from this search query, while 14 and 87 publications were retrieved from ScienceDirect and Google Scholar respectively. After removing duplicates and those with the irrelevant abstract contents, 15 publications were selected for further analysis.

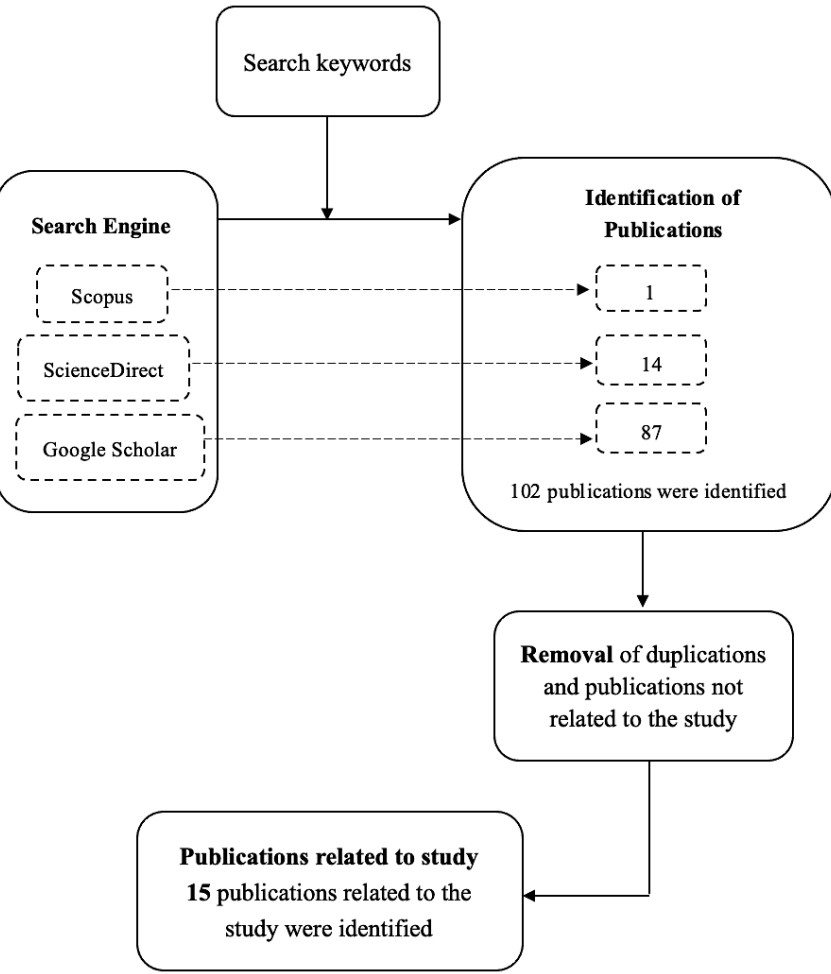

**Figure 2.** An overview of the literature review process for stage 2.

## 3. The Status of 3D Concrete Printing in the Construction Industry

### 3.1. Benefits and Construction Applications

The method of 3D concrete printing (3DCP) has been increasingly used in the construction industry in recent years where concrete is the main construction material employed [13]. Khoshnevis proposed Contour Crafting (CC) in the early 1990s, which is a workable system for 3DCP and particularly applicable for large building structures [23]. Current 3D concrete printing processes include binder jetting, material extrusion [24], and robotic shotcrete [25]. Binder jetting is a process according to which the material particles are selectively joined using a binder, while the material extrusion used is suitable for the conventional materials. The process extrudes the materials on a predefined path that solidifies later (layer by layer) [4]. Shotcrete shows higher mechanical properties and enhanced interlayer bonding than extrusion-based printing, but also faces many challenges such as precision of the control, as well as pumpability and shootability of the material.

Although it is too early to say whether 3D concrete printing can completely replace the current concrete construction methods such as cast-in-situ and pre-fabrication, many studies have argued that this emerging technology can be used to eliminate some of the complexities and negative effects of current construction methods [2,26,27]. Currently, the concrete construction industry faces scores of challenges, including the cost of formwork, the physical labor involved in setting up the formwork and placing the rebar, and the limitations on the shape of the structure [11,26]. Formwork is an important component of in situ concrete construction projects, accounting for 35–50% of the total cost of a concrete structure and about 50–75% of the total construction time [28]. Complex projects accentuate the significant negative impact of formwork on time and the environment [29]. The method of 3D printing minimizes the need for formwork use and labor involvement [30], thereby reducing costs [31], eliminating many construction hazards [32], and reducing environmental hazards such as waste and noise pollution during construction [33]. In addition, freedom of design and improved quality are also important benefits that 3D printing can bring. Without the limitations of formwork, 3DCP can easily print curved walls and other structural elements, allowing architects to conceive complex geometries that go beyond traditional rectilinear design concepts [2]. Digitization is another advantage of 3D printing, meaning that designs can be converted into printer instructions, which minimizes the unnecessary errors and waste that may arise from interpreting drawings during construction, as well as site operation, hence enhancing the quality.

The construction industry has a high environmental impact worldwide, and 3D printing is expected to mitigate the environmental impact of traditional construction by reducing energy use, resource demand and $CO_2$ emissions over the product lifecycle [34]. A study consisting of designing a two-story building by different construction methods found that 3DCP was the most economical and sustainable method, reducing construction time by approximately 95% (not considering prefabrication), providing the greatest cost savings, and generating approximately 32% less $CO_2$ emissions [35].

Figure 3 compares conventional construction (CC) and prefabricated construction (PC) relative to 3DCP in terms of the total weight of the structure, construction time, final cost, and total $CO_2$ emissions. It shows that 3D printing is not only more sustainable and environmentally friendly than the other two construction methods but also has significant advantages in respect of construction time and cost. These advantages also highlight its promising potential for affordable housing for low-income people, and for local housing reconstruction after natural disasters like earthquakes, hurricanes, and floods [4].

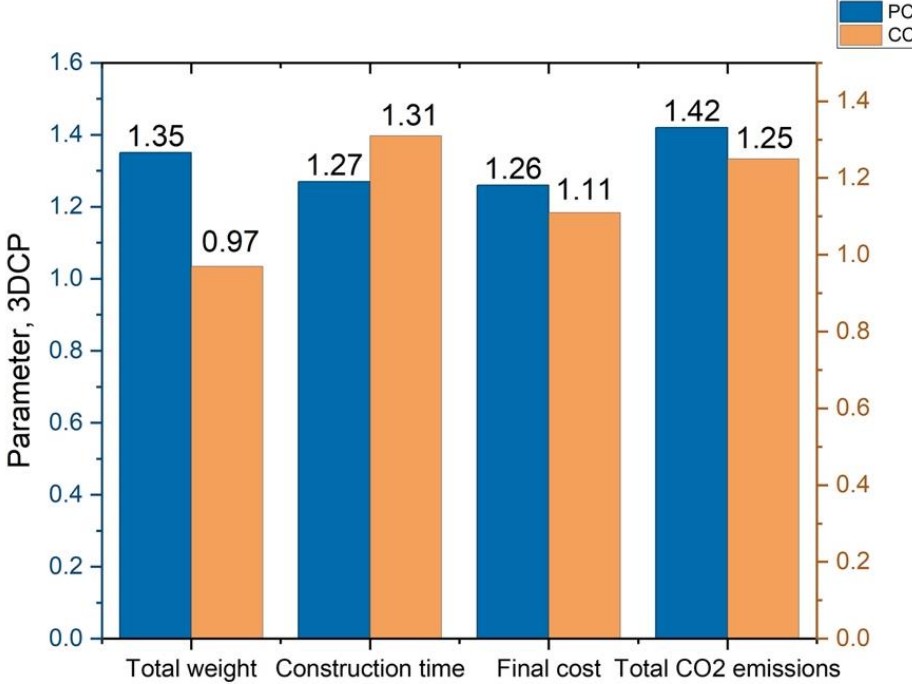

**Figure 3.** The relative factor between PC, CC and 3DCP [35–37].

As the construction industry becomes increasingly interested in 3D printing, more and more projects using this technology are emerging and revealing its benefits and promises. Until 2014, 3DCP was mostly used in small projects including landscape architecture, street furniture, and non-structural components, with few large-scale demonstration projects being implemented. After that, the number of projects employing 3DCP in housing has increased dramatically [10]. The use of 3DCP in the project of the farmhouse in Wujiazhuang village saved 62.4% of labor cost and reduced on-site construction time by 24.5% compared to traditional reinforced concrete (RC) methods [38]. The advantage of 3DCP is that it succeeds in bringing more flexibility and efficiency, as well as environmental benefits, when employed in printing structures with complex designs and right-angle walls [4]. In addition to single building structures, this technology has also shown its superiority in large-scale construction. WinSun has successfully printed multi-unit houses in a short period of time and at a low cost ($4800 USD per unit), showing the promise of 3D printing in building low-cost houses [6]. The company also built an exhibition hall in 2021 with a total area of about 2400 square meters consisting of 3D printed modular components which demonstrates the feasibility of using 3DCP in combination with prefabricated assemblies for large-scale construction [10]. Furthermore, the development of 3D printing is not limited to Earth as it is increasingly being considered as a means of building space habitats, especially a viable option for building a permanent base on the Moon because of the advantage of being able to use in situ resources [5,39].

### 3.2. Barriers and Challenges

Using 3D printing is becoming increasingly popular and enjoys many benefits, but as any new technology, it is not perfect [1]. Barriers and challenges can also become important issues as they could point to the direction of technological development and accelerate its growth.

The method of 3DCP is typically suitable for single-story structures, as concrete slabs poured on-site are necessary for multi-story construction projects. In addition, the larger the structure to be printed, the larger the 3D printer needed, and thus the less accurate it will be. Therefore, the major challenge is how to print tall and complex structures. Considering that 3D printing is more limited in terms of the size of the structure, printing multi-story

buildings requires a combination of prefabrication and assembly techniques [10], which means that each floor will be printed in a factory and then transported to the site for floor-by-floor installation which can bring another challenge: the lack of proper external support and the removal of it after construction [40].

Reinforcement is important for the bearing capacity and robustness of the structure, but remains one of the challenges of 3D concrete printing [41]. There are currently two main solutions to this challenge. The first one is to manually place the reinforcement between the layers before or during printing, but this method increases the difficulty of moving the print head as well as the labor cost [2]. The other is the use of fiber-reinforced concrete as a printing material, which enhances structural stability by increasing the strength and ductility of the printed part [1]. Chemical additives also play an important role in regulating the properties of the print concrete, such as setting time, fluidity, and mechanical strength, etc. [4]. Therefore, the development of appropriate materials that meet both printability and acceptable mechanical properties is also an important challenge for 3DCP [42].

The layer-by-layer appearance is one of the unavoidable features of 3DCP given the nature of additive manufacturing [13]. In the case of special application scenarios, such as emergency housing, where the need for speed and scale of construction is much higher than the appearance, a layer-by-layer appearance is acceptable. However, for consumer dwellings, some chemical or physical post-treatment methods like sintering are important to mitigate this defect since a flat appearance of architecture is more aesthetically pleasing to the public [12,43]. Dimensional errors and improper control of deposited material (insufficient or excessive) are the main factors that lead to 3DCP buildings having poor surfaces, so the printing speed and material output are required to be controlled, otherwise, expensive surface post-treatment needs to be considered [44,45].

The lack of information on the long-term durability and longevity of 3D printed structures is also a challenge; as 3DCP is a relatively new concept, the duration of its earliest applications is still very short and not all materials used are standardized. Some features of 3DCP such as the lack of templates to prevent air exposure may lead to accelerated evaporation of moisture from the printed structure, therefore increasing the risk of shrinkage and cracking [13]. It also remains to be determined whether the durability, stiffness, strength, load-bearing limit values and other important indicators of their materials can meet the code requirements within the construction industry due to a lack of specific standardization. Therefore, further research to explore and develop guidelines and standards, as well as more refined and targeted maintenance methods are urgently needed [46]. In addition, 3D concrete printing requires attention to the environment in which it works, as the properties of the concrete materials needed for printing can be affected by a range of environmental factors including dust, temperature, and humidity.

## 4. Digital Twin Application in 3DCP

*4.1. Overview*

The complexity of 3D printing technology is evident as it is a multidisciplinary and cross-border integrated technology system, including the digitization of building models, structural design, concrete materials, intelligent printing systems, and assembly technology. Considering the use of digital technologies in 3DCP, some of its barriers and limitations, including the large investment in R&D, the complexity of quality control, and the lack of life-cycle information management, can be expected to be improved by integrating advanced digital twin (DT) technologies. In general, DT refers to the virtual twin representation of the objects or systems using the best available data, and so far a typical application of the DT concept in the construction industry would be the building information modeling (BIM) technique [47]. BIM involves creation and the use of a model (digital twin) for managing the digital information of the buildings and extending traditional 3D modeling as it provides seamless integration and management of the entire lifecycle, including scheduling (the fourth dimension, so-to-speak), cost estimating (the fifth dimension) [48,49], and sustainability (carbon emission), leading to a 6D representation of a construction

process. The method of 6D BIM was investigated as a method to guide large construction projects (e.g., railway stations), optimizing the construction process, sustainability, and logistics through efficient information management [50]. The digital twin is essentially a transparent information hub, providing digital resources for the design, manufacturing, construction, operation and maintenance, demolition, and recycling of the architectures (i.e., the life cycle of buildings). It plays a key role in sharing information among engineers, project managers, and technicians in different life cycle stages of the buildings, meaning that different stakeholders can communicate technical complexities and arrive at a smart decision, hopefully improving the sustainability and reducing the risk and cost of the buildings. The digital twin (BIM model) has been designed for the lifecycle management of different systems including social housing [51], railway turnover system [52], and subway system [50]. The integration of the BIM models has dramatically lowered carbon emissions and capital costs over the whole lifecycle of systems. However, the potential of DT in lifecycle management for 3DCP systems has not been well studied. Current research on the use of digital twins in 3DCP is fragmented, with some publications just mentioning their expectations of integrating DT technology to improve 3D printing and their proposed application directions for the different stages of 3DCP in a few sections.

### 4.2. Applications and Benefits in Different Lifecycle Stages

Currently, DT is rarely considered and applied in the construction industry [22], and the concept of DT and its capabilities have not been distinguished from the commonly used computational or virtual models and simulations [53]. Unlike static BIM, DT is expected to be "self-aware" and self-optimizing to enable dynamic two-way conversation and control [54]. DT facilitates the connection of the physical environment and digital ecosystem through many self-operating features that are expected to further enhance the performance and sustainability of 3D printing.

Figure 4 shows an interactive system of physical assets with a data model that explores integrated digital delivery across whole lifecycles. The digital models enable engineers to validate designs and accurately determine and address defects in manufactured components (3D printed modules, etc.), thereby improving the quality and performance of the project. The application and role of digital twin technology in 3D printing projects vary in different lifecycles, and the direction of DT application was discussed below in each of the three critical lifecycle stages.

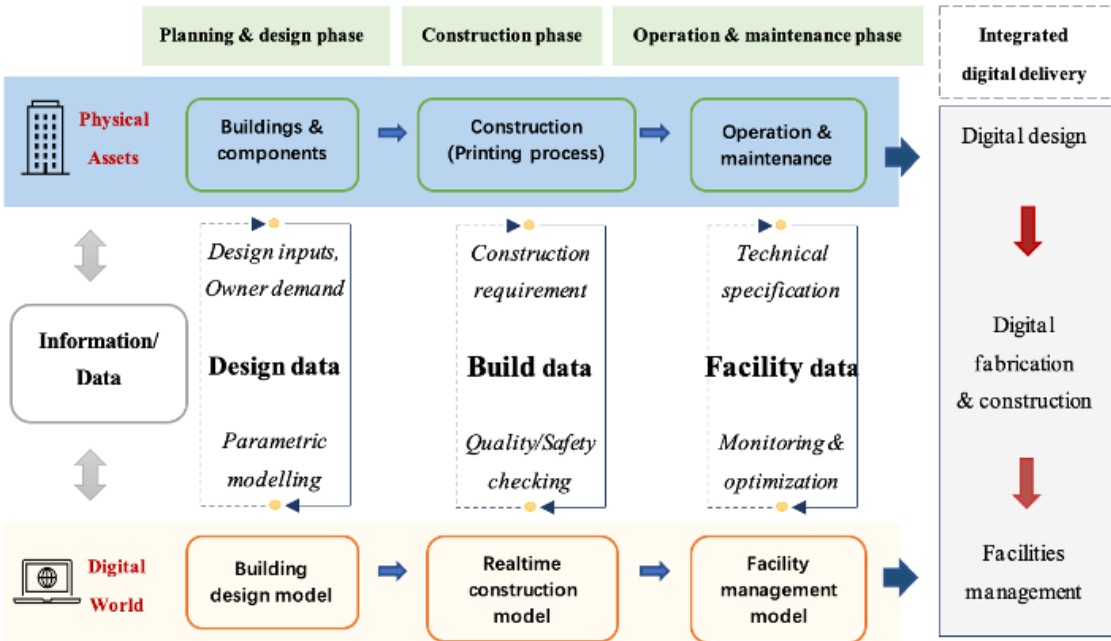

**Figure 4.** Integrated digital delivery across the whole life cycle of buildings (adapted from Ref. [55]).

### 4.2.1. Planning and Design Phase

BIM integrates various kinds of key data, such as dimensions, material management, equipment utilization, resources, etc., which does not only facilitates communication and cooperation between different stakeholders but also allows for a higher level of automation in the planning process [56]. Information sharing and updating on the BIM platform can effectively minimize the lead time; for example, design modifications can be made directly in 3D printing, reducing unnecessary intermediate processes [57]. In addition, BIM as a data integration plays many other important roles in the initial phase. Different expertise, such as engineering knowledge, can be embedded in the BIM platform to help architects to improve design feasibility and reduce modification costs in the initial design process [58]. Detailed information about each project will be stored in the BIM database for future projects or as a base for further design, supporting decision-making regarding material selection, energy management, procurement, and the like. This feature is especially useful in large-scale or modular 3D printing projects, where the high reusability of digital data (databases as well as algorithms, etc.) reduces unit costs and minimizes planning costs. Moreover, adding wireless sensor networks (WSN) to BIM creates dynamic real-time models that embody DT concepts, providing designers with effective information to make informed decisions during the project design process [59]. The digital twin can be used to study the structural behavior during the 3D printing process [60] to reduce the amount of trial and error testing and defects and therefore reduce design time [61]. It could also effectively reduce the cost of diagnostic tests by creating a DT model to simulate 3DCP structures [62].

### 4.2.2. Construction Phase

BIM excels at storing and arranging material delivery data, printer control data, and post-finishing operations data, facilitating automation in 3D concrete printing [30,49]. The method of 5D BIM integrates project quantity information, progress information, and cost information, which can both evaluate the project cost and enable managers to better control the project schedule and cost inputs by monitoring the breakdown progress during the construction period and using digital models to simulate the construction process. The use of various equipment including large printers, conveying pipes, and pumping machines impose certain requirements on the site safety of 3DCP projects. In this case, DT can come into play to achieve human-machine interaction for intelligent safety solutions through real-time location sensing (RTLS) and two-way conversation between the physical and digital environments [44]. The digital twin allows the development of a real-time control system that combines physical space and cyberspace so that problems that occur during the printing process like fluctuations in material properties or pumping would be immediately identified and responded to without interrupting the process; this greatly improves its robustness [63]. The DT model can initiate the correct actions for the upcoming printing process based on real-time data sets and it is considered to be effective in improving the automation of 3DCP as well as machine learning techniques [64].

DT is necessary for 3D printing modularization projects, where real-time networking of products, processes, and systems can increase the efficiency of modularization efforts and reduce waste. In addition, DT can be used to assist various management activities, including resource management, material management, schedule management, quality management, etc., to try to follow up on the construction progress and ensure project quality [22] and safety [61].

### 4.2.3. Operation and Maintenance Phase

In this phase, the project goes through different stakeholders, which causes difficulties in integrating data between different phases and stakeholders. Subsequently, DT can be used as a platform to enhance the flow of information between different stakeholders for facility and maintenance management, monitoring, and energy simulation of the project. Digital twin technology aids analysis and decision-making by collecting real-time data to manage building operations and maintenance and building energy consumption [21]. The life-cycle performance of 3D printing technology has yet to be determined and requires more attention during the construction and maintenance phases, especially if the use of materials and construction methods differ from the traditional ones [13]. By using information from monitoring devices such as sensors installed in the structure, DT can provide effective means to monitor the comfort level and maintenance needs of the structure, improving sustainability and refining the long-term performance data of 3D-printed buildings.

The demolition and recovery phases are usually bypassed in the research work. To the best of the authors' knowledge, DT applications for these phases (especially for 3DCP projects) have not been studied in the existing literature, which represents a research gap.

The different lifecycle phases have demonstrated that digital twin can have a significant impact on 3D printing projects, providing the opportunity to effectively address challenges before they occur. Overall, DT-based 3D printing will be beneficial with respect to quality improvement, cost and labor savings, and stakeholders of construction projects will benefit from the application of DT. However, DT is still in its infancy in the construction industry and in 3D printing projects, and more research is needed to bridge the gap between 3D printing and digital twin.

## 5. Discussion

As mentioned in Section 3, the four main challenges of current 3D concrete printing in the construction industry were investigated: limitations of the size of the structures, the robustness of the structure, layer-by-layer appearance, and lack of standardization. The corresponding potential solutions were also summarized, and it was found that many of them could be optimized using digital twin technology.

Existing research presents potential applications and benefits of digital twins in 3D concrete printing from different lifecycles, as described in Section 4. The current research on the use of digital twins in 3DCP is quite scarce and the ideas proposed are scattered and fragmented. Through reviewing and summarizing, the applications of DT have been divided into three main lifecycle phases: planning and design phase, construction phase and operation and maintenance phase. In the initial stages, DT can optimize the design and reduce the trial and error costs through the integration of information and real-time model simulations. During the construction phase, real-time bi-directional information interaction improves the accuracy of scheduling and control of printing systems on site, reducing problems and disruptions. In the maintenance phase, DT can be primarily used to monitor the status of the structure and share the information for long-term analysis and maintenance. Figure 5 describes how the digital twin can be used at three different lifecycle stages to optimize or solve the challenges that 3D concrete printing faces, and shows targeted directions and pathways for further research.

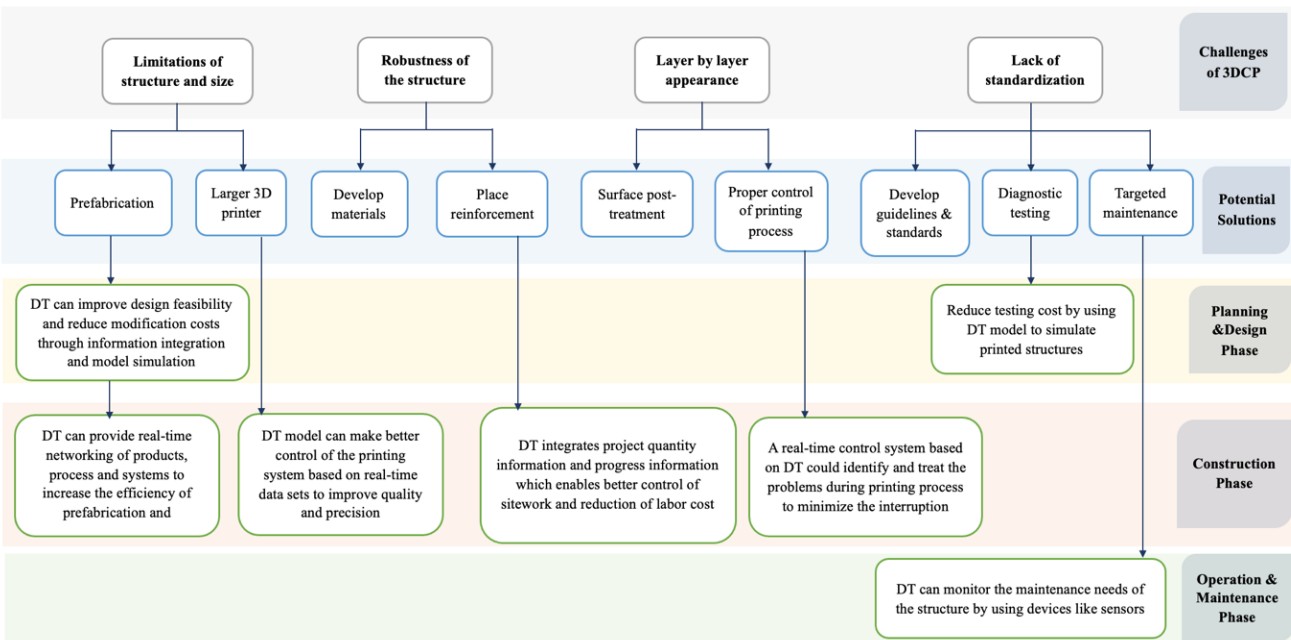

**Figure 5.** DT applications in response to 3DCP challenges.

## 6. Conclusions

The method of 3D concrete printing technology and digital twin are both technologies and concepts that have been popular in the construction industry recently. Given that 3D printing projects face scores of challenges and barriers to development, including scale limitations, high initial investment, environmental sensitivities, complexity in quality control, and lack of lifecycle management, the concept of a digital twin (DT) is being considered for introduction into the full lifecycle of 3D printing projects, utilizing different application methods to function at various stages. Specific applications and future directions of DT in different stages of 3D printing projects were discussed. It was found that the validation of its mechanism of action and specific effects is needed. The applications focus on the planning to the operation and maintenance phases. On the one hand, DT can form an information platform to facilitate the communication and cooperation of various stakeholders and to promote construction management and database management. Furthermore, it provides a two-way data exchange, linking physical reality and virtual models, to monitor and observe the development of the project at all stages in a targeted manner and thus make early response decisions. The study also presents a roadmap detailing the main challenges that 3D concrete printing faces and how potential solutions can be optimized by introducing the technology of the digital twin which provides potential directions and pathways for future research to develop 3DCP. Future research could explore the design of 3DCP project cases incorporating DT to demonstrate its effectiveness in reducing challenges and improving quality and sustainability as well as its application in non-standard situations such as extraterrestrial construction.

**Funding:** This research received no external funding.

**Institutional Review Board Statement:** Not applicable.

**Informed Consent Statement:** Not applicable.

**Data Availability Statement:** Not applicable.

**Acknowledgments:** The first author acknowledges the support from the University of Western Australia International Fee Scholarship and an Australian Government Research Training Program (RTP) Scholarship.

**Conflicts of Interest:** The authors declare no conflict of interest.

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
