# Peer review of "Digital Twin Applications in 3D Concrete Printing"

_sustainability, doi:10.3390/su15032124_

Round 1

Reviewer 1 Report

The concept of the paper is quite interesting and relevant to the objectives of the journal of sustainability.

Here are some comments to improve the quality of the paper.

·       Add a summary about three main concrete 3D printing including binder jetting, robotic shotcrete, and layered material extrusion.

·       Add some details regarding adding reinforcement and additives to concrete 3D printing technology.

Author Response

The concept of the paper is quite interesting and relevant to the objectives of the journal of sustainability.

Here are some comments to improve the quality of the paper.

  • Add a summary about three main concrete 3D printing including binder jetting, robotic shotcrete, and layered material extrusion.

  • Add some details regarding adding reinforcement and additives to concrete 3D printing technology.

Thank you so much for the comments and suggestions. A summary of three main concrete 3D printing processes (Line 122) and some details regarding adding reinforcement and additives to 3DCP (Line 205) have been added to the manuscript.

Reviewer 2 Report

- Abstract is too short, need to add more detail of review article

- Caption of Fig-2 is not representing true meaning

- Data for the article is not sufficient, discussion on topic is non-focused, probably lack the rigor due to deficient of detailed figures/charts/graphs.

Author Response

Reviewer2

- Abstract is too short, need to add more detail of review article

- Caption of Fig-2 is not representing true meaning

- Data for the article is not sufficient, discussion on topic is non-focused, probably lack the rigor due to deficient of detailed figures/charts/graphs.

Thank you very much for the comments and suggestions. Responses are as follows:

  1. Abstract has been updated by adding more details, identification of challenges and the findings.
  2. The caption has changed to “Performance comparison of PC, CC and 3DCP” (Line 168)
  3. A discussion section and more data and figures have been added. (Section 4&5)

Reviewer 3 Report

The MS titled "Digital twin applications in 3D concrete printing" is more like a summary than a scientific contribution to the 3DPC field. This MS is not suitable to be published in Sustainability (MDPI) based on the following comments:

1. No line numbers and section numbers are provided that made the manuscript very difficult to review.

2. The abstract is too short with no proper identification of problem/challenges, solutions, and justification to be published.

3. In many places, lump references are used. The author should consider eliminating those multiple references. This should be done by characterizing each reference individually by mentioning 1 or 2 phrases per reference to show how it is different from the others and why it deserves mentioning. Multiple references are of no use for a reader and can substitute even a kind of plagiarism, as sometimes authors are using them without proper studies of all references used. In the case, each reference should be justified by providing short assessment.

4. Aims and objectives are not provided at the end of introduction section. 

5. The methodology section is missing, which represent any standard methodology adopted for the review article and databases used. 

6. A review article is not the summary of previously published articles but critical analysis of available data. On the basis of such analysis, authors should provide their own contribution, which leads to new dimensions in the respective field. This aspect is completely missing from this MS. 

Author Response

Reviewer3

The MS titled "Digital twin applications in 3D concrete printing" is more like a summary than a scientific contribution to the 3DPC field. This MS is not suitable to be published in Sustainability (MDPI) based on the following comments:

  1. No line numbers and section numbers are provided that made the manuscript very difficult to review.

  1. The abstract is too short with no proper identification of problem/challenges, solutions, and justification to be published.

  1. In many places, lump references are used. The author should consider eliminating those multiple references. This should be done by characterizing each reference individually by mentioning 1 or 2 phrases per reference to show how it is different from the others and why it deserves mentioning. Multiple references are of no use for a reader and can substitute even a kind of plagiarism, as sometimes authors are using them without proper studies of all references used. In the case, each reference should be justified by providing short assessment.

  1. Aims and objectives are not provided at the end of introduction section.

  1. The methodology section is missing, which represent any standard methodology adopted for the review article and databases used.

  1. A review article is not the summary of previously published articles but critical analysis of available data. On the basis of such analysis, authors should provide their own contribution, which leads to new dimensions in the respective field. This aspect is completely missing from this MS.

Thank you so much for the detailed comments and suggestions. Responses are as follows:

  1. Line numbers and section numbers have been added.
  2. Abstract has been updated by adding more details, identification of challenges and the findings.
  3. I apologize for not separating multiple references in some places before. All the references have been checked and many of the references that were together have been separated and placed after the corresponding phrases and sentences to show more clearly where they were cited from.
  4. The aims and objectives of the paper have been added at the end of the introduction section (Line 80).
  5. The methodology section and a figure showing the process have been added (line 92)
  6. Section 4 has been updated by adding more data and a discussion section (section 5) has been added to show the findings and a roadmap describing the main challenges 3D concrete printing faces and how potential solutions can be optimized by introducing the technology of the digital twin which provides potential directions and pathways for future research to develop 3DCP.

Reviewer 4 Report

the proposed system is suited for 3 DP mass production, however due to specific interlocking design , which is basically a good idea, the system needs special edge, corner, lintel etc blocks to be widely buildable.

Author Response

Reviewer4

the proposed system is suited for 3 DP mass production, however due to specific interlocking design , which is basically a good idea, the system needs special edge, corner, lintel etc blocks to be widely buildable.

Thank you very much for the comments and it has been updated in line 64.

Round 2

Reviewer 3 Report

Can be accepted as most of comments are addressed.